# Ethnolichenology—The Use of Lichens in the Himalayas and Southwestern Parts of China

**Mei-Xia Yang [1,2,*], Shiva Devkota [3,4], Li-Song Wang [5] and Christoph Scheidegger [1,2,*]**

1   Swiss Federal Institute for Forest, Snow and Landscape Research WSL, 8903 Birmensdorf, Switzerland
2   Faculty of Sciences, University of Bern, 3012 Bern, Switzerland
3   Global Institute for Interdisciplinary Studies (GIIS), Kathmandu 3084, Nepal; shiva.devkota@gmail.com
4   Himalayan Climate & Science Institute (HCSI), Washington, DC 20007, USA
5   Key Laboratory for Plant Diversity and Biogeography of East Asia, Kunming Institute of Botany, CAS, Kunming 650201, China; wanglisong@mail.kib.ac.cn
*   Correspondence: meixia124@gmail.com (M.-X.Y.); christoph.scheidegger@wsl.ch (C.S.); Tel.: +41-79-836-8441 (M.-X.Y.); +41-79-460-7132 (C.S.)

**Abstract:** Lichens are used in traditional medicine, food and various other ethnic uses by cultures across the Himalayas and southwestern parts of China. Evidence-based knowledge from historical and modern literatures and investigation of ethnic uses from 1990 proved that lichen species used as medicine in the Himalayas and southwestern parts of China totaled to 142 species; furthermore, 42 species were utilized as food. Moreover, some lichens are popularly used for lichen produce in ethnic and modern life. An understanding and clarification of the use of lichens in the Himalayas and southeastern parts of China can therefore be important for understanding uses of lichens elsewhere and a reference for additional research of lichen uses in the future.

**Keywords:** lichen; ethnic use; medicinal; edible species; Himalayas; southwestern China

## 1. General Introduction of Lichen Uses

Lichens are composite organisms containing algae (e.g., *Trebouxia* or *Trentepohlia*), or cyanobacteria (*Nostoc*), living among filaments of multiple fungal species in a mutualistic relationship [1,2]. Lichens dominate vegetation types on about 7% of the planet's surface; additionally, they are important components of primary producers in a wide range of substrates and habitats, including some of the most extreme conditions on earth (North and South Pole, desert, even glass surfaces, etc.) [3]. Many lichens (such as *Usnea* Dill. ex Adans., *Everni* Ach., *Hypogymnia* (Nyl.) Nyl., *Parmelia* Ach. et al.) are very sensitive to environmental disturbances and they can be used to assess air pollution [4–6]. Unlike simple dehydration in plants and animals, lichens may experience a complete loss of body water in dry periods [7]. Lichens are important in contributing nitrogen to soils either by forming litter, or predation by herbivores, e.g., snails, which then defecate, providing nitrogen to the soils [8]. In deserts and semi-arid areas, lichens are part of extensive, living biological soil crusts, essential for maintaining the soil structure. They have a long fossil record in soils, dating back 2.2 billion years [9].

Lichens are also important in diets for humans and animals. Based on research on the diet of *Rhinopithecus roxellana* Milne-Edwards in China, lichens are the most eaten food for *Rh. roxellana* (Figure 1a), accounting for 38.4% of the overall diet [10]. The regional diversity of lichens will also affect the changes in the living area of *Rh. roxellana*. Moreover, for the human diet, mostly in the temperate and arctic regions of the world, people usually use lichens as food, pharmaceutical products, and various ethnic uses [11]. In the past, Iceland moss (*Cetraria islandica* (L.) Ach.) was an important source of food for humans in northern Europe and the lichen was cooked as bread, porridge, pudding, soup, or salad.

Wila (*Bryoria fremontii* (Tuck.) Brodo & D. Hawksw.) was an important food in parts of North America, where it was usually pit cooked. Northern peoples in North America and Siberia traditionally eat the partially digested reindeer lichen (*Cladonia* P. Browne) after they remove it from the rumen of caribou or reindeer that have been killed. Rock tripe (*Umbilicaria* Hoffm. and *Lasallia* Mérat) is a lichen (sometimes more species of lichens) that has frequently been used as an emergency food in North America and one species, *Umbilicaria esculenta* (Miyoshi) Minks, is used in a variety of traditional Korean and Japanese foods [12].

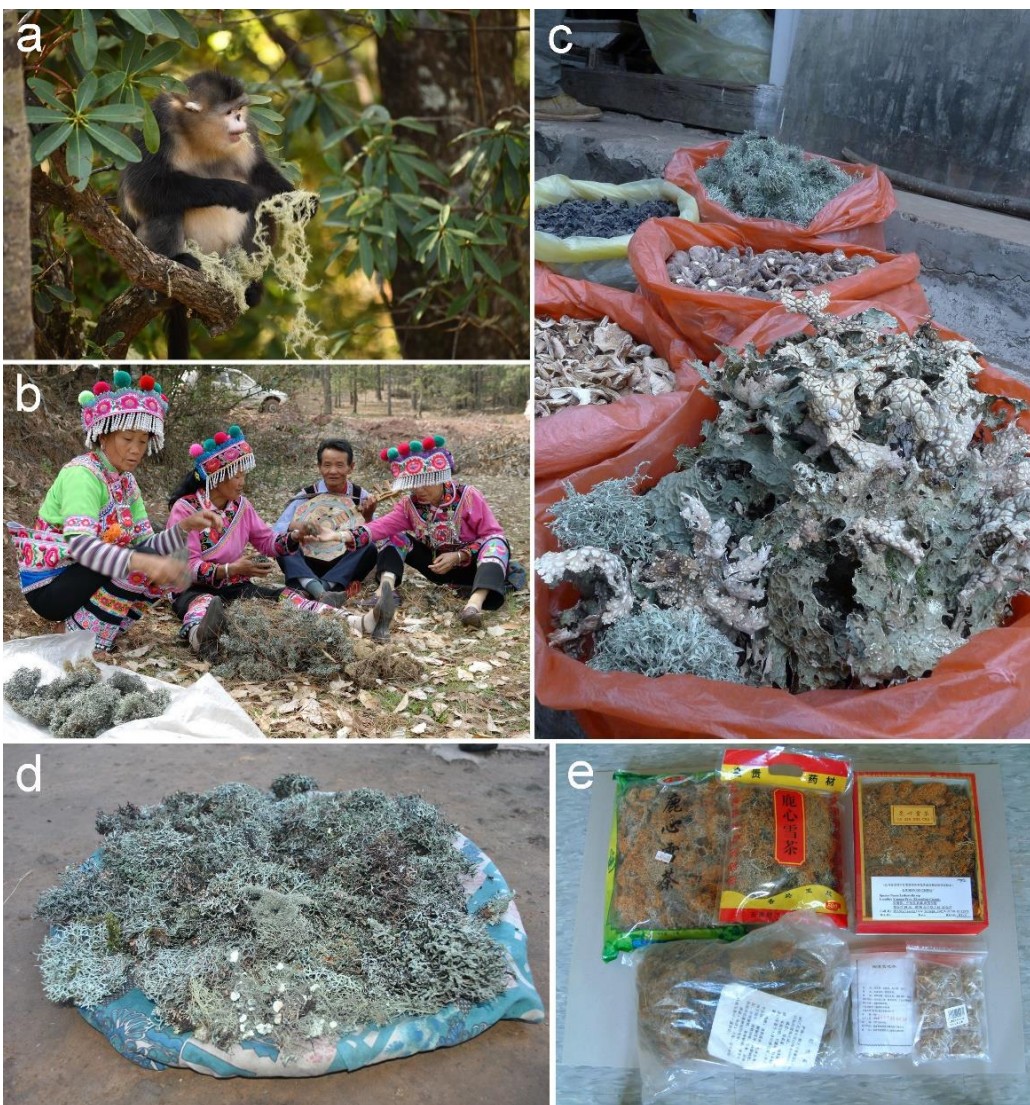

**Figure 1.** Lichens are used by animals and humans. (**a**) *Rhinopithecus roxellana* eating *Usnea* in southwestern China. (**b**) Bai minority people harvesting lichens in Yunnan. (**c**) Ethnic lichen market in Yunnan. (**d**) Freshly gathered lichens for the kitchen in Nepal. (**e**) Health-promoting tea products of *Lethariella* and *Thamnolia*. (**a–c,e**) photographed by Li-Song Wang; (**d**) photographed by Shiva Devkota.

Ethnolichenology is a branch of ethnobotany that studies the uses that man makes of lichens traditionally [13,14]. Lichens are used for many different medicinal purposes, but there are some general categories of use that reoccur across the world. Lichens are often drunk as a decoction to treat ailments relating to either the lungs or the digestive system [15–18]. This is particularly common in the Himalayas and southeastern China. Many other uses of lichens are related to treating gynecological diseases. This may be related to the common use of lichens for treating sexually transmitted infections and aliments of the urinary system [11]. Two other uses of lichens that are less common, but reoccur in several

different cultures, are the treatment of eye afflictions and use in smoking mixtures [11]. Besides, lichens are often used externally for dressing wounds, either as a disinfectant or to stop bleeding [18]. Other common topical lichen uses are for skin infections and sores, including sores in the mouth [11,19]. Many of the traditional medicinal uses of lichens are probably related to their secondary metabolites, many of which are known to both be physiologically active and act as antibiotics [15]. However, some of the traditional uses of lichens also rely on the qualities of lichen carbohydrates. Many of the traditional uses of lichens involve boiling the lichen to create a mucilage which is drunk for lung or digestive ailments, or applied topically for other issues [17,18,20]. Other lichen carbohydrates which may be important are the isolichenins and galactomannans, which are widespread across various taxonomic groups of lichens, and the pustulins, that are found in Umbilicariaceae [11].

Today, ethnic groups inhabiting the mighty Himalayas (Bhutan, China, India and Nepal) primarily adapt the classical systems of medicine following Ayurveda, Siddha, Unani, Traditional Chinese Medicine (TCM) and Amchi practices and continue their traditional uses of lichens for food, beverages and traditional medicine. Within the last decade, however, the sale of lichens for folk uses, especially for supposedly health-promoting teas, has increased remarkably, as the Himalayas and southwestern parts of China have become popular regions for domestic tourism (Figure 1b,e).

Since 1990, the authors have studied these folk uses as part of a broader investigation of the lichen flora of southwestern parts of China, with aims to reveal the species diversity of lichens used traditionally and currently, the diversity of various uses by studying records from herbaria and other notes and interviewing current population. The present paper summarizes the results of our ethnobotanical investigation, which has already been treated in our previous studies [21–26], and other related research references, which are listed in this study. We expect that the understanding of the use of lichens in the Himalayas and southeastern parts of China will serve as an important reference for additional research of lichen uses.

## 2. Materials and Methods

Chinese samples are available in the Lichen Herbarium of the Kunming Institute of Botany (KUN-L) and we reviewed useful lichen collections housed at different herbaria (KATH, TUCH) and at the Natural History Museum, Tribhuvun University, Nepal, for their additional notes, if any. We interviewed local people about their uses of lichens and visited ethnic markets to buy samples of the lichens offered for sale and also to ensure the legitimacy, relevance and credibility of the given evidence.

Specimens were examined using standard microscopic techniques and hand-sectioned under a Nikon SMZ 745T dissecting microscope. Anatomical descriptions are based on observations of these preparations under a Nikon Eclipse 50i microscope. Secondary metabolites of all the specimens were identified using spot tests and thin-layer chromatography (TLC), as described by White and James [27] and Orange et al. [28]. Solvent system C (toluene: acetic acid = 85:15) was used for TLC analysis.

## 3. Results

### 3.1. Traditional Medicinal Lichens in the Himalayas and Southwestern Parts of China

Introduction of Typical Medicinal Lichens

Some of the typical, traditionally applied lichens are presented in Figure 2. Several species of *Lobaria* (Schreb.) Hoffm. are frequently used traditional medicines effective for treating pneumonia in the Himalayas and southwestern China, due to their lung-like appearance (applied because of the doctrine of signatures, suggesting that herbs can treat body parts that they physically resemble) [23,29]. *Lobaria* species have also been reported to serve as a valuable source of proteins, having a protein content higher than that of kelp or edible fungi, such as *Tremella* Pers. In addition, the content of dietary fiber in *Lobaria*

species is significantly higher than in other fungi and edible algae and *Lobaria* species are rich in calcium [30]. Similarly, *Peltigera leucophlebia* (Nyl.) Gyeln. is used as a supposed cure for thrush (Aphtha, Candidiasis), due to the resemblance of its cephalodia to the appearance of the disease [31]. The earliest report about traditional medicin of lichen could be the *Usnea longissima* Ach. in the Chinese Qing dynasty (ca. 1500); it was reported that the *Usnea* species were used as litmus, to treat cold, swelling and pain [32]. Lichens also have their place in current pharmaceutical research; lichens produce metabolites of potential therapeutic or diagnostic value [18]. Some metabolites produced by lichens are structurally and functionally similar to broad-spectrum antibiotics, while a few of them are associated, respectively, with antiseptic similarities [33]. Usnic acid is the most commonly studied metabolite produced by lichens [15]. It is also under research as a bactericidal agent against *Escherichia coli* (Migula) Castellani & Chalmers and *Staphylococcus aureus* Bergey.

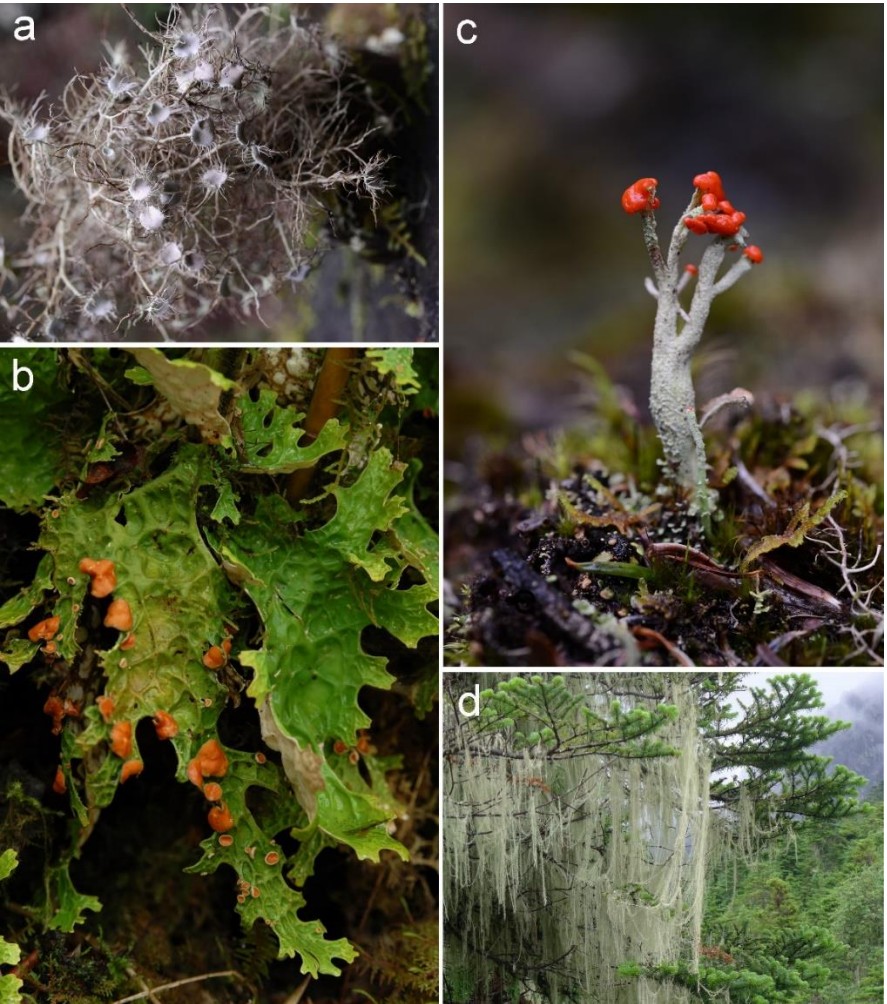

**Figure 2.** Selected traditional medicinal lichens. (**a**) *Sulcaria sulcata* (Lév.) Bystrek. (**b**) *Cladonia* sp. (**c**) *Lobaria* sp. (**d**) *Usnea longissima*. Photographed by Li-Song Wang.

### 3.2. Summary of Research on Medicinal Lichens

Textual research of several publications have documented the inventory and ethnic uses of lichens from Nepal [22,34–37], India [20–41], Bhutan [19,42] and southwestern parts of China, in ancient [32] and modern research [14,23,25,43,44]. The literature and investigation of folk usages proved that lichen species used as medicine in the Himalayas and southwestern parts of China totaled to 142 species belonging to 16 families and 46 genera. Figure 3 provides the species number within the different genera of traditional

medicinal lichens. Table 1 presents lichen species in alphabetical order and provides the details on each traditional use.

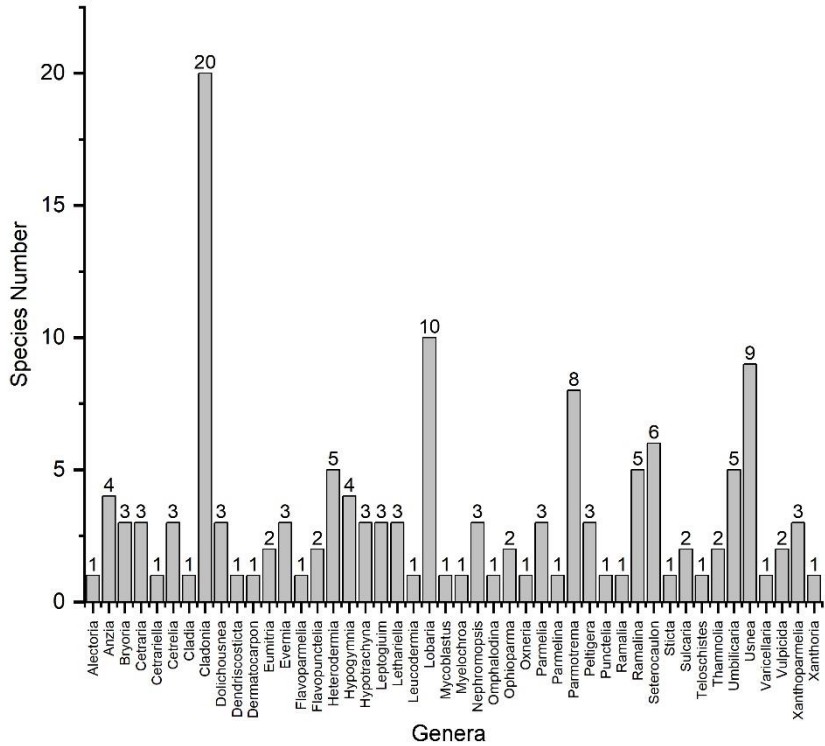

**Figure 3.** Species number within the different genera of traditional medicinal lichen.

**Table 1.** Lichen species used in traditional medicine or herbal medicine in the Himalayas and southwestern parts of China. The newly added ones through our investigation are indicated in boldface.

| Current Species Names | Function and Traditional Application | Folk Names | Main Area of Use | References |
|---|---|---|---|---|
| *Alectoria ochroleuca* (Schrank) A. Massal. | a, b | | China | [16,43] |
| *Anzia formosana* Asahina | b | | China | [43] |
| *Anzia japonica* (Tuck.) Müll. Arg. | a | | China | [43] |
| *Anzia opuntiella* Müll. Arg. | a | | China | [29] |
| *Anzia ornata* (Zahlbr.) Asahina | a | | China | [29,43] |
| ***Bryoria asiatica* (Du Rietz) Brodo & D. Hawksw.** | l | 树发 (*Tree hair*) | China | [23] |
| ***Bryoria bicolor* (Hoffm.) Brodo & D. Hawksw.** | l | 树发 (*Tree hair*) | China | [23] |
| ***Bryoria confusa* (D.D. Awasthi) Brodo & D. Hawksw.** | l | 树发 (*Tree hair*) | China | [23] |
| *Cetraria* sp. | | | India | [17,45] |
| ***Cetraria islandica* (L.) Ach.** | a | | China | [23] |
| ***Cetraria laevigata* Rass.** | a | | China | [23] |
| *Cetrariella delisei* (Bory ex Schaer.) Kärnefelt & A. Thell | a | | China | [43] |
| *Cetrelia cetrarioides* (Delise) W.L. Culb. & C.F. Culb. | a, b | | China | [43] |

| | | | | |
|---|---|---|---|---|
| *Cetrelia olivetorum* (Nyl.) W.L. Culb. & C.F. Culb. | a, b | | China | [43] |
| *Cetrelia pseudolivetorum* (Asahina) W.L. Culb. & C.F. Culb. | a | | China | [29] |
| *Cladia aggregata* (Sw.) Nyl. | b | 石花菜 (*Stone flower*) | China | [29,43] |
| *Cladonia amaurocraea* (Flörke) Schaer. | a, b | 青雪茶 (*Green snow-tea*) | China | [16] |
| *Cladonia arbuscula* (Wallr.) Flot. | l | | China | [16] |
| *Cladonia bellidiflora* (Ach.) Schaer. | a | | China | [16] |
| *Cladonia cenotea* (Ach.) Schaer. | b | | China | [29,43] |
| *Cladonia cervicornis* (Ach.) Flot. | c | | China | [16,46] |
| *Cladonia crispata* (Ach.) Flot. | o | | India | [17,20] |
| *Cladonia cyanipes* (Sommerf.) Nyl. | a | | China | [29,43] |
| *Cladonia digitata* (L.) Hoffm. | d | | China | [16] |
| **Cladonia fenestralis** **Nuno** | l | | China | [23] |
| *Cladonia fruticulosa* Kremp. | b, d | | China | [16,43] |
| *Cladonia floerkeana* (Fr.) Flörke | a | | China | [43] |
| **Cladonia gracilis** **(L.) Willd.** | c | 太白鹿角 (*Taibai antlers*) | China | [23,47,48] |
| *Cladonia macilenta* Hoffm. | a, b | | China | [43] |
| *Cladonia macroceras* (Delise) Ahti | l | | China | [29] |
| *Cladonia mitis* Sandst. | a | | China | [16,43] |
| *Cladonia pleurota* (Flörke) Schaer. | a | | China | [29,43] |
| *Cladonia pyxidata* (L.) Hoffm. | b | | China | [43] |
| *Cladonia rangiferina* (L.) Weber | b | | China | [29] |
| *Cladonia squamosa* (Scop.) Hoffm. | b | | China | [29,43] |
| **Cladonia stellaris** **(Opiz) Pouzar & Vězda** | a | 太白花 (*Tai-bai flower*) | China | [23,29,43] |
| **Dolichousnea diffracta** **(Vain.) Articus** (as *Usnea diffracta* Vain.) | k | 老君须 (*Lao Jun's beard*) | China | [23,47] |
| **Dolichousnea longissima** **(Ach.) Articus** (as *Usnea longissima* Ach.) | a, n, q | 松萝 | China, India | [18,23,38] |
| *Dolichousnea trichodeoides* (Vain. ex Motyka) Articus (as *Usnea trichodeoides* Vain.) | a, b | | China | [29] |
| *Dendriscosticta wrightii* (Tuck.) B. Moncada & Lücking (as *Sticta wrightii* Tuck.) | d | | China | [46,49] |
| *Dermatocarpon miniatum* (L.) W. Mann | d, e | | China | [50] |
| *Eumitria baileyi* Stirt. (as *Usnea baileyi* (Stirt.) Zahlbr.) | m | | India | [18] |
| *Eumitria pectinata* (Taylor) Articus (as *Usnea pectinata* Taylor) | a, b | | China | [29] |
| *Evernia divaricata* (L.) Ach. | a, f | | China | [29,43] |
| *Evernia esorediosa* (Müll. Arg.) Du Rietz | a, f, g | | China | [29,43] |
| *Evernia mesomorpha* Nyl. | a, f | | China | [29,43] |
| *Flavoparmelia caperata* (L.) Hale | a | | China | [16,29] |
| *Flavopunctelia flaventior* (Stirt.) Hale | a, b | | China | [16,43] |
| *Flavopunctelia soredica* (Nyl.) Hale | a, b | | China | [43] |
| *Heterodermia comosa* (Eschw.) Follmann & Redón | b | | China | [16,43] |

| | | | | |
|---|---|---|---|---|
| *Heterodermia diademata* **(Taylor) D.D. Awasthi** | b,d | झुलो<br>(*Jhulo*) | China, India, Nepal | [16,17,22,43] |
| *Heterodermia hypochraea* (Vain.) Swinscow & Krog | b | | China | [29,43] |
| *Heterodermia pseudospeciosa* (Kurok.) W.L. Culb. | b | | China | [29,43] |
| *Heterodermia speciosa* (Wulfen) Trevis. | b | | China | [16] |
| *Hypogymnia flavida* McCune & Obermayer | a | | China | [29,43] |
| *Hypogymnia hypotrypa* (Nyl.) Rass. | a | | China | [29,43] |
| *Hypogymnia physodes* (L.) Nyl. | b | | China | [29,43] |
| *Hypogymnia pseudoenteromorpha* M.J. Lai | b | | China | [43] |
| *Hypotrachyna cirrhata* **(Fr.) Divakar, A. Crespo, Sipman, Elix & Lumbsch** (as *Everniastrum cirrhatum* (Fr.) Hale) | f | | China | [23,51,52] |
| *Hypotrachyna nepalensis* **(Taylor) Divakar, A. Crespo, Sipman, Elix & Lumbsch** | a, f | इ्याउ<br>(*Jhyauu*) | China, Nepal | [16,23,37] |
| *Hypotrachyna sinuosa* (Sm.) Hale | a | | China | [29,43] |
| *Leptogium delavayi* **Hue** | l | | China | [23] |
| *Leptogium saturninum* **(Dicks.) Nyl.** | l | | China | [23] |
| *Leptogium trichophorum* **Müll. Arg.** | l | | China | [23] |
| *Lethariella cladonioides* **(Nyl.) Krog** | b | 红雪茶<br>(*Red snow-tea*) | China | [24,29] |
| *Lethariella flexuosa* **(Nyl.) J.C. Wei** | h | 红雪茶<br>(*Red snow-tea*) | China | [24] |
| *Lethariella zahlbruckneri* **(Du Rietz) Krog** | h | 红雪茶<br>(*Red snow-tea*) | China | [24,44] |
| *Leucodermia boryi* (Fée) Kalb (as *Heterodermia boryi* (Fée) Kr.P. Singh & S.R. Singh) | b | | China | [16,43] |
| *Lobaria sp* | o | | Bhutan | [19] |
| *Lobaria isidiosa* (Müll. Arg.) Vain. | e | 老龙皮<br>(*Dragon skin*) | China | [23,29] |
| *Lobaria kurokawae* Yoshim. | e | 树蝴蝶<br>(*Tree butterfly*) | China | [29] |
| *Lobaria meridionalis* Vain. | e | 树蝴蝶<br>(*Tree butterfly*) | China | [29] |
| *Lobaria orientalis* **(Asahina) Yoshim.** | l | 树蝴蝶<br>(*Tree butterfly*) | China, India | [17,23,45] |
| *Lobaria pindarensis* **Räsänen** | l | 树蝴蝶<br>(*Tree butterfly*) | China | [23] |
| *Lobaria pulmonaria* (L.) Hoffm. | d | 蛤蟆七<br>(*Toad skin*) | China | [17,48] |
| *Lobaria retigera* **(Bory) Trevis.** | d, i | 老龙皮<br>(*Dragon skin*) | China | [23,30] |
| *Lobaria sublaevis* (Nyl.) Yoshim. | d | 树蝴蝶<br>(*Tree butterfly*) | China | [43] |
| *Lobaria yunnanensis* **Yoshim.** | l | 树蝴蝶<br>(*Tree butterfly*) | China | [23] |
| *Mycoblastus alpinus* (Fr.) Th. Fr. ex Hellb. | c | | China | [16,46,50] |
| *Myelochroa irrugans* (Nyl.) Elix & Hale | b | | China | [24] |

| | | | | |
|---|---|---|---|---|
| *Nephromopsis cucullata* (Bellardi) Divakar, A. Crespo & Lumbsch (as *Flavocetraria cucullata* (Bellardi) Kärnefelt & A. Thell) | a | | China | [43] |
| *Nephromopsis nivalis* (L.) Divakar, A. Crespo & Lumbsch (as *Flavocetraria nivalis* (L.) Kärnefelt & A. Thell) | a | | China | [29] |
| *Nephromopsis pallescens* (Schaer.) Y.S. Park | c | | China | [23] |
| *Omphalodina chrysoleuca* (Sm.) S.Y. Kondr., L. Lőkös & Farkas (as *Rhizoplaca chrysoleuca* (Sm.) Zopf) | d | | China | [16] |
| **Ophioparma lapponica (Räsänen) Hafellner & R.W. Rogers** | c | | China | [23,29] |
| *Ophioparma ventosa* **(L.) Norman** | c | | China | [23,29] |
| *Oxneria fallax* (Arnold) S.Y. Kondr. & Kärnefelt (as *Xanthoria fallax* Arnold) | b | | China | [16,29] |
| *Parmelia adaugescens* Nyl. | b | | China | [16] |
| *Parmelia saxatilis* (L.) Ach. | a, b | 石花 (*Rock flower*) | China | [23,29,43] |
| *Parmelia sulcata* Taylor | b | | China | [29,43] |
| *Parmelina quercina* (Willd.) Hale | b | | China | [29,43] |
| *Parmelinella wallichiana* (Taylor) Elix & Hale (as *Parmelina wallichiana* (Taylor) Hale) | b | | China | [43] |
| *Parmotrema abessinicum* (Nyl. ex Kremp.) Hale | a | Rathipuvvu (*Rock flower*) | India | [17] |
| *Parmotrema cetratum* (Ach.) Hale (as *Rimelia cetrata* (Ach.) Hale & A. Fletcher) | b | | China | [16] |
| *Parmotrema chinense* (Osbeck) Hale & Ahti | a | Chharila | India | [45] |
| *Parmotrema nilgherrense* (Nyl.) Hale | a, d, e, l | | India | [45] |
| *Parmotrema reticulatum* (Taylor) M. Choisy (as *Rimelia reticulata* (Taylor) Hale & A. Fletcher) | b | | China | [29,43] |
| *Parmotrema sancti-angelii* (Lynge) Hale | p | Jhau | India | [17,20,38] |
| *Parmotrema subtinctorium* (Zahlbr.) Hale | c | | China | [16] |
| *Parmotrema tinctorum* (Despr. ex Nyl.) Hale | a, b | | China | [29,43] |
| *Peltigera aphthosa* (L.) Willd. | d | | China | [16,29] |
| *Peltigera canina* (L.) Willd. | j | | China | [16,45] |
| *Peltigera polydactylon* (Neck.) Hoffm. | l | | China, India | [16,17,29] |
| *Punctelia borreri* (Turner) Krog | a, b | | China | [16] |
| ***Ramalia sp*** | d | Jhyauu (*Unnecessary stuff*) | Nepal | [22] |
| *Ramalina commixta* Asahina | f | 石花菜 (*Stone flower*) | China | [51,52] |
| *Ramalina conduplicans* Vain. | d | 石花菜 (*Stone flower*) | China | [16] |
| *Ramalina fastigiata* (Pers.) Ach. | a, f | 石花菜 (*Stone flower*) | China | [29,43] |
| *Ramalina roesleri* (Schaer.) Nyl. | b, f | 石花菜 (*Stone flower*) | China | [29,43] |
| *Ramalina sinensis* Jatta | a, f | 石花菜 (*Stone flower*) | China | [29,43] |
| *Stereocaulon exutum* Nyl. | c | | China | [29] |

| | | | | |
|---|---|---|---|---|
| *Stereocaulon himalayense* D.D. Awasthi & I.M. Lamb | | Dhungo-ku-Jhau (*Rock flower*) | India | [17] |
| **Stereocaulon japonicum Th. Fr.** | b, d | | China | [23] |
| **Stereocaulon myriocarpum Th. Fr.** | b | | China | [23] |
| **Stereocaulon paschale (L.) Hoffm.** | b | 石寄生 (*Rock parasite*) | China | [23,29] |
| *Stereocaulon tomentosum* Fr. | b, d | | China | [23,43] |
| *Sticta gracilis* (Müll. Arg.) Zahlbr. | | | India | [17] |
| *Sulcaria sulcata* (Lév.) Bystrek | f, h | 石花菜 (*Stone flower*) | China | [43,51,52] |
| **Sulcaria virens (Gyeln.) Bystrek** | c | | China | [23] |
| *Teloschistes flavicans* (Sw.) Norman | l | | China | [16] |
| **Thamnolia subuliformis (Ehrh.) W.L. Culb.** | k | 白雪茶 (*White snow-tea*) | China | [23,24,43] |
| **Thamnolia vermicularis (Sw.) Schaer.** | k | 白雪茶 (*White snow-tea*) | China | [23,24,32,43] |
| **Umbilicaria esculenta (Miyoshi) Minks** | d, h | 石耳 (*Stone ear*) | China | [23,29,43] |
| *Umbilicaria hypococcinea* (Jatta) Llano | a | | China | [16,29] |
| *Umbilicaria nanella* Frey & Poelt | d | | China | [48,53] |
| *Umbilicaria vellea* (L.) Ach. | l | | China | [16] |
| **Umbilicaria yunnana (Nyl.) Hue** | l | | China | [23] |
| *Usnea aciculifera* Vain. | h | | China | [16] |
| *Usnea ceratina* Ach. | k | | China | [16] |
| *Usnea florida* (L.) F.H. Wigg. | a, c | | China | [43,47,50] |
| *Usnea nidifica* Taylor | l | | China | [16] |
| *Usnea rubicunda* Stirt. | a | | China | [29,43] |
| *Usnea rubrotincta* Stirt. | | | India | [17,20] |
| **Usnea subfloridana Stirt.** | a | | China | [23] |
| *Usnea subsordida* Stirt. | r | | India | [19,54] |
| *Usnea* spp. | a, d, i, n | | India, Bhutan | [17,18,20,42] |
| **Varicellaria velata (Turner) I. Schmitt & Lumbsch** (as *Pertusaria velata* (Turner) Nyl.) | c | | China | [23,29] |
| *Vulpicida juniperinus* (L.) J.-E. Mattsson & M.J. Lai | a | | China | [43] |
| *Vulpicida pinastri* (Scop.) J.-E. Mattsson & M.J. Lai | a | | China | [43] |
| *Xanthoparmelia camtschadalis* (Ach.) Hale | a | | China | [16,29] |
| *Xanthoparmelia taractica* (Kremp.) Hale | a | | China | [29] |
| *Xanthoparmelia tinctina* (Maheu & A. Gillet) Hale | a | | China | [29] |
| *Xanthoria parietina* (L.) Th. Fr. | b, d | | China | [16,29,43] |

The abbreviations indicate different functions and traditional applications: "a", raw material for antibiotics; "b", raw material for making litmus reagent; "c", hemostatic of external injury; "d", anti-inflammatory and antibacterial; "e", digestion facilitator and stomach enhancer; "f", raw material for making spice; "g", antihypertensive; "h", anti-colon cancer; "i", calcium level booster; "j", treatment of rabies and icterus; "k", detoxifying and cough suppressant; "l", other medicinal; "m", mixed with other aromatic herbs, such as *Valeriana jatamansi*, for flavor and curing tobacco; "n", against lung troubles, hemorrhages and asthma attacks, strengthening of hair, treatment of skin eruptions and boils, stopping nose bleeds, preventing or treating blisters; "o", anti-tumor; "p", against skin diseases; "q", treatment of bone fracture; "r", flavoring tobacco.

### 3.3. Edible Lichens in the Himalayas and Southwestern Parts of China

3.3.1. Introduction of Typically Edible Lichen

In the past, edible lichens were mostly gathered for private consumption [24]. In recent years, edible lichens are increasingly sold by local people or tourists as a commodity, after drying in the Himalayas and southwestern parts of China (Figure 4). *Lethariella* (Motyka) Krog and *Thamnolia* Ach. ex Schaer. are widely used as health-promoting teas, *Lobaria*, *Umbilicaria* Hoffm., *Nephromopsis* Müll. Arg. and *Ramalina* Ach. are used as food and are relatively common in the local supermarkets and some restaurants. Normally, summer and autumn are the best seasons to harvest edible lichens, which are used fresh or dried for later use. Usually, stewing with burns, steaming, cooking soup and other methods are used to make dishes, such as "Liang Ban" *Ramalina*, etc.

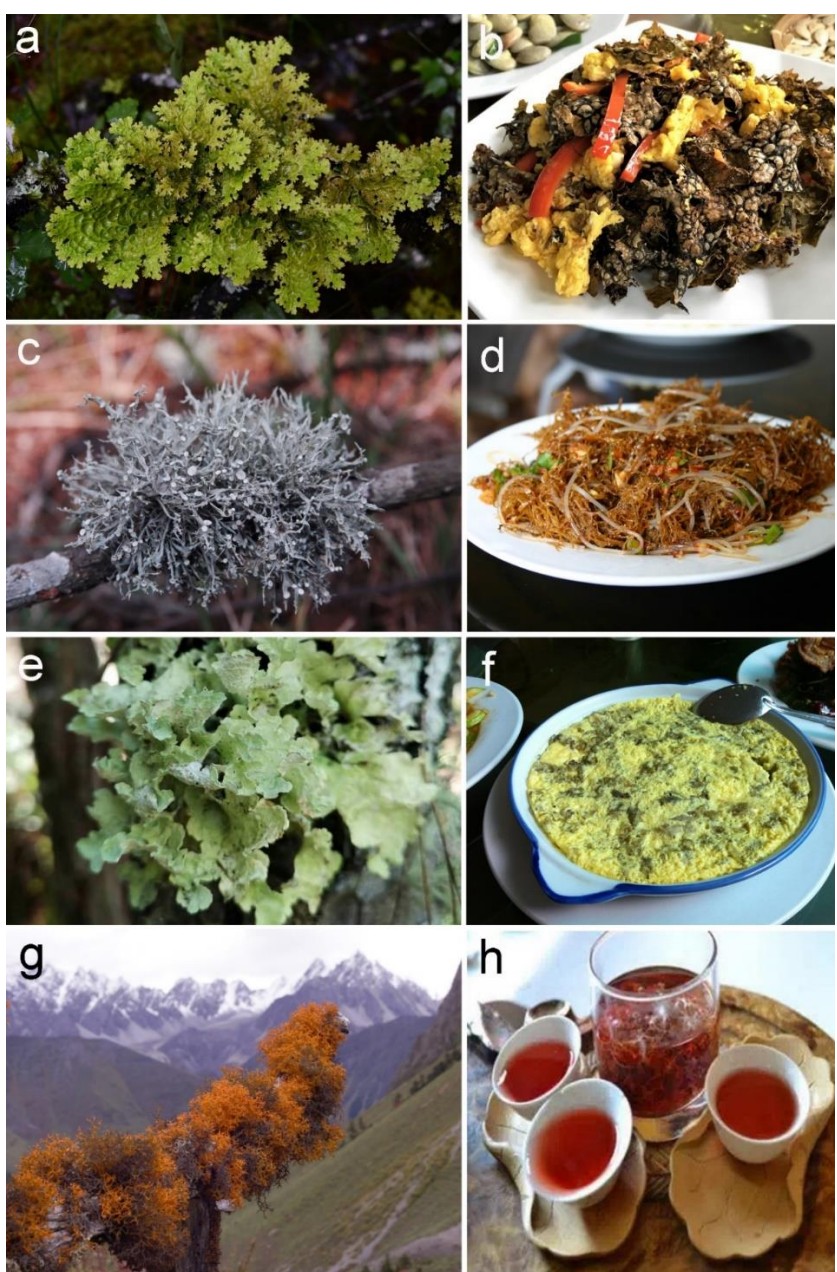

**Figure 4.** Selected edible lichens in the Himalayas and southwestern China. (**a**) *Lobaria*. (**b**) Scrambled eggs with *Lobaria*. (**c**) *Ramalina fastigiata*. (**d**) "Liang Ban" *Ramalina*. (**e**) *Nephromopsis pallescens* (Schaer.) Y.S. Park. (**f**) Egg custard with *Nephromopsis pallescens*. (**g**) *Lethariella*. (**h**) *Lethariella* tea. Photographed by Li-Song Wang.

3.3.2. Summary of Research on Edible Lichen

By researching ancient and modern literature and investigating folk usages in the Himalayas and southwestern parts of China, it is obvious that people generally name edible lichen "shuhua" or "shihuacai"; the names mean flowering form of an organism growing on the trees or stones and these phrases are still in use. Here we document a total of 42 species belonging to 18 genera of lichens that are edible (Table 2). The most commonly used genera of lichens are foliose or fruticose growth forms (such as *Hypotrachyna*, *Lethariella*, *Lobaria*, *Nephromopsis*, *Ramalina*, *Thamnolia* and *Umbilicaria*). The edible lichens counted in this article are limited and more investigation and research are needed in the future.

**Table 2.** Edible lichens in Himalayas and southwestern China.

| Consumed | Species Name |
|---|---|
| | *Bryoria asiatica* (Du Rietz) Brodo & D. Hawksw. |
| | *Bryoria confusa* (D.D. Awasthi) Brodo & D. Hawksw. |
| | *Cetraria laevigata* Rass. |
| | *Cladia aggregata* (Sw.) Nyl. |
| | *Cladonia gracilis* (L.) Willd. |
| | *Cladonia stellaris* (Opiz) Pouzar & Vězda |
| | *Dermatocarpon miniatum* (L.) W. Mann |
| | *Hypotrachyna cirrhata* (Fr.) Divakar, A. Crespo, Sipman, Elix & Lumbsch |
| | *Hypotrachyna nepalensis* (Taylor) Divakar, A. Crespo, Sipman, Elix & Lumbsch |
| | *Leptogium delavayi* Hue |
| | *Leptogium saturninum* (Dicks.) Nyl. |
| | *Leptogium trichophorum* Müll. Arg. |
| | *Leptogium wilsonii* Zahlbr. (as *Leptogium denticulatum* Nyl.) |
| (Eaten) as vegetable, with different cooking methods, such as stewing, steaming, boiling, frying, making soup, etc. [17,18,21–24,26,30,37] | *Leucodermia leucomelos* (L.) Kalb (as *Heterodermia leucomelos* (L.) Poelt) |
| | *Lobaria isidiophora* Yoshim. |
| | *Lobaria isidiosa* (Müll. Arg.) Vain. |
| | *Lobaria kurokawae* Yoshim. |
| | *Lobaria orientalis* (Asahina) Yoshim. |
| | *Lobaria pindarensis* Räsänen |
| | *Lobaria pulmonaria* (L.) Hoffm. |
| | *Lobaria retigera* (Bory) Trevis. |
| | *Lobaria yunnanensis* Yoshim. |
| | *Nephromopsis pallescens* (Schaer.) Y.S. Park |
| | *Parmotrema cetratum* (Ach.) Hale |
| | *Parmotrema reticulatum* (Taylor) M. Choisy |
| | *Parmotrema tinctorum* (Despr. ex Nyl.) Hale |
| | *Ramalina commixta* Asahina |
| | *Ramalina conduplicans* Vain. |
| | *Ramalina fastigiata* (Pers.) Ach. |
| | *Ramalina roesleri* (Schaer.) Nyl. |
| | *Ramalina sinensis* Jatta |
| | *Sulcaria sulcata* (Lév.) Bystrek |

| | |
|---|---|
| | *Umbilicaria esculenta* (Miyoshi) Minks |
| | *Umbilicaria hypococcinea* (Jatta) Llano |
| | *Umbilicaria yunnana* (Nyl.) Hue |
| | *Usnea longissima* Ach. |
| | *Cladonia fenestralis* Nuno |
| | *Lethariella cladonioides* (Nyl.) Krog |
| (Drunk) as tea [24,26,30] | *Lethariella flexuosa* (Nyl.) J.C. Wei |
| | *Lethariella zahlbruckneri* (Du Rietz) Krog |
| | *Thamnolia subuliformis* (Ehrh.) W.L. Culb. |
| | *Thamnolia vermicularis* (Sw.) Schaer. |

## 4. Other Ethnic and Modern Uses

Lichens are mainly used by humans for medicine and foods in the Himalayas and southwestern parts of China, but we have also found many other, novel uses for these organisms in a local place. *Lethariella*, only distributed at around 3700–4300 m in the Himalayas, is mainly sold in Yunnan and Shangri-La, China, and also exported to Taiwan and Japan. In the Himalayas, Tibet is a holy place of Buddhism and there is a great demand for Tibetan incense; *Lethariella* is also used as an important component of Tibetan incense because of its special fragrance (Figure 5a). Besides, in current applications of lichens, *Usnea* and *Sulcaria* are used for raw materials of perfume and fragrance in Yunnan (Figure 5b). *Cladonia* is commonly used as garden decoration in China. Devkota et al. [22] reported ritual and spiritual value (RSV), aesthetic and decorative value (ADV), bedding value (BV) and ethno-veterinary values (EVV) of lichens, together with medicinal value (MV) and food value (FV), among different collectors and indigenous people and local communities (IPLCs) in Nepal. *Cetrelia collata* (Nyl.) W.L. Culb. & C.F. Culb. (current name: *Platysma collatum* Nyl.) is used as a sacrificial fiber, together with *Melanelia infumata* (Nyl.) Essl., *Everniastrum cirrhatum* and *Parmotrema nilgherrense*, *Usnea ghattensis* G. Awasthi, for coloring hair, *Thamnolia vermicularis* (Sw.) Schaer., with its spiritual value in India and Nepal [22,45], and *Buellia subsororioides*, used to color palms and lips as a substitute for Heena, mostly by the Garhwali Herdsman in Uttarakhand and India [39]. Furthermore, Shukla et al. [55] have also highlighted the use of eleven lichen species as dying agents in Gharwal region of India.

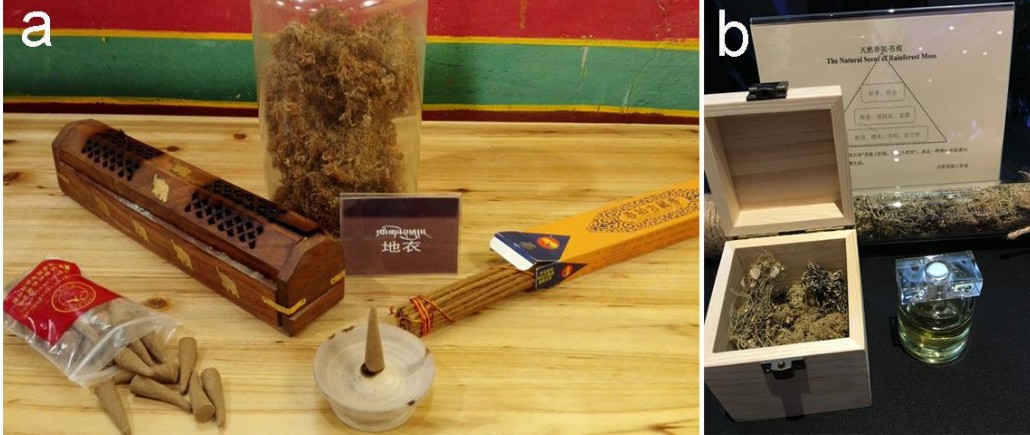

**Figure 5.** (**a**) *Lethariella*, used for Tibetan incense. (**b**) *Usnea* and *Sulcaria*, used for raw materials of perfume in Yunnan. (**a**) Photographed by Mei-Xia Yang; (**b**) photographed by Li-Song Wang.

## 5. Discussion and Conclusions

Our investigation and survey of the literature indicate that 142 lichen species are used as medicine and 42 species are used as food in the Himalayas and southwestern parts of

China. We found considerable overlap between the medicinal and consumed lichens; except for three species of edible lichens (*Leptogium wilsonii*, *Leucodermia leucomelos* and *Lobaria isidiophora*), other species with edible uses also have medicinal functions (Tables 1 and 2). Therefore, the popularity of consuming healthy food items might be explained by their preventive role. Almost all food lichens are cooked in some way before being eaten and the cooking process is often complex, usually involving steps to remove toxins from the lichen [11,22,23,25,43]. For the medicinal lichens, the secondary compounds and carbohydrates are useful to humans. The studies reported that the nutritionally relevant carbohydrates in lichens include the glucans lichenin and isolichenin [11,14]. Some lichens also have significant levels of proteins and essential amino acids, as well as some minerals and vitamins, but most lichens only have minimal amounts of these nutrients [11,12,15,30]. Some lichens are only eaten in times of famine, some are a staple food or even a delicacy in the Himalayas and southwestern China. The medicinal and edible usage of lichen as healthy food is becoming more and more popular among local people and tourists. However, two obstacles are often encountered when eating lichens: lichen polysaccharides are generally indigestible to humans and lichens usually contain mildly toxic secondary compounds that should be removed before eating. Very few lichens are poisonous, but those having high concentrations of vulpinic acid or usnic acid are toxic [56].

Lichen resources are especially abundant in the Himalayas and southeastern parts of China and it is also a major advantage that they mostly grow at high altitudes without human activity and pollution. In recent years, resource survey and research on lichens as food and medicine have also been reported in these areas. Unfortunately, there are still some problems concerning the classification of species and the unclear distribution of resources; therefore, for example, the species classification of the genus *Lobaria* and the resource distribution of *Lobaria* in these areas also need to be investigated. At present, most lichens for food and traditional medicine are used directly as lichen raw materials and, considering that many lichens grow slowly, it results that lichens productivity is usually low. However, there are still some ecosystems of foliose lichens (e.g., *Lobaria pulmonaria*) that can produce significant lichen biomass within approximately a decade [57,58]. The low productivity of lichens means that over-harvesting is a real concern. For example, the genus *Lethariella*—with known distribution only in the Himalayas and southeastern parts of China—has reached an endangered state before its active ingredients and mechanisms could be fully understood. Based on our recent study, we expect to carry out further relevant research on effective medicinal and nutritional ingredients of lichens in the future and explore ways to obtain the required effective ingredients through artificial culturing or by fermentation, also to reduce the dependence on natural resources. Therefore, combining results of lichen taxonomy, ecology, chemistry and pharmacology is a top priority for our forthcoming research. At the same time, effective protection measures for some endangered species are important for the sustainable use of lichen resources.

**Author Contributions:** Conceptualization, C.S. and M.-X.Y.; methodology, M.-X.Y.; software, M.-X.Y.; validation, M.-X.Y. and C.S.; formal analysis, M.-X.Y.; investigation, L.-S.W., C.S. and S.D.; resources, L.-S.W. and S.D.; data curation, M.-X.Y., L.-S.W. and S.D.; writing—original draft preparation, M.-X.Y.; writing—review and editing, M.-X.Y., C.S., S.D. and L.-S.W.; visualization, M.-X.Y. and C.S.; supervision, C.S.; project administration, C.S.; funding acquisition, C.S., S.D. and L.-S.W. All authors have read and agreed to the published version of the manuscript.

**Funding:** This work was supported by the Swiss National Science Foundation (grant JRP IZ70Z0_131338/1 to CS), the National Natural Science Foundation of China (No. 31970022, 31670028), the Second Tibetan Plateau Scientific Expedition and Research Program (No. 2019QZKK0503), the Global Biodiversity Information Facility/Biodiversity Fund for Asia (Project No: BIFA5_023 to SD) and the China Scholarship Council (CSC No. 201704910901).

**Institutional Review Board Statement:** Not applicable.

**Data Availability Statement:** The data that support the findings of this study are available from the corresponding author, upon reasonable request.

**Acknowledgments:** We sincerely thank Xin-Yu Wang (Kunming Institute of Botany, CAS, China) and Dong Liu (Central South University of Forestry and Technology, China), for supporting expedition and research in China; the Lichen Herbarium of the Kunming Institute of Botany, CAS for providing research support; Ram Prasad Chaudhary (RECAST Kathmandu, Nepal) and Krishna Kumar Shrestha (Tribhuvan Univ. Kathmandu, Nepal), for supporting research in Nepal; and Karma Tshering, for giving insights into ethnobotany in Bhutan.

**Conflicts of Interest:** The authors declare no conflict of interest.

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
