# Peer review of "Ethnolichenology—The Use of Lichens in the Himalayas and Southwestern Parts of China"

_diversity, doi:10.3390/d13070330_

Round 1
Reviewer 1 Report
This work reviewed for the first time the use of lichens in the Himalayas and southwestern parts of China from a medical to a culinary point of view. I found this work very interesting and, moreover, well-written. Nowadays, this kind of topic has a good scientific resonance since they are able to connect scientific to cultural uses. In my opinion, this work has raised an important concern related to the use of lichens as “food” or basis for drug development, i.e. the one connected to their increasing use (due to the local tourism) which when associated with their slow growth can generate excessive harvesting of these ecological precious organisms. In fact, this is why, since the Journal is called “Diversity” I would see also some information (or a small paragraph) about the impact of the harvesting on lichen biodiversity, in order to better integrate the work with the selected Journal. Every part of the article is well-constructed and informative and I’m not able to find negative points on its structure.
The author is requested to check only a small error: Line 64: anaimal
Reviewer 2 Report
The paper “Ethnolichenology - the Use of Lichens in the Himalayas and Southwestern Parts of China” is a well written manuscript presenting review of study in scope of ethnical using of lichens in traditional medicine and as a food by mainly the people Himalayas and southwestern parts of China. In my opinion, it is very interesting because, this article provides an overview of research based on data from 1990 to present, according to study of folk uses as a part of a broader lichens flora researches of the China (SW). Authors reviewed lichen herbariums (KATH, TUCH and the Natural History Museum, Tribhuvun University, Nepal) for their additional notes and they interviewed local people about their uses of lichens and visited ethnic markets to buy samples of the lichens to credibility of the given evidence.
[lines 61-62] “The present paper summarizes the results of our ethnobotanical investigation and research reference” - I think it should be made clear which data comes from the literature and which are the authors' own data.
In general assessment – technical quality and clarity of presentation are very good. I think, the importance of this work is excellent.
Title is accurately reflects the content of the paper.
Language: I think, is grammatically good and appropriate and understandable, but I don't feel qualified to judge about the English language and style detail.
Abstract – There are no information what's the knowledge gap that you're trying to fill and why did you want to go about this review in the abstract. What was the general kind of approach - your own study (scope of time), literature sources, that you took in order to understand this problem?
Keywords: are informative and adequate.
Introduction: are clearly presented, adequately connected with the state-of-knowledge, but in my opinion there is no definition of the term ethnolichenology and lack of clear information which was the aim of this work. I think it is possible to write a few sentences about the role of lichen secondary metabolites in the context of the article too.
Discussion and Conclusions I don't know if the discussion should be a separate part of the paper. In fact, the discussion is conducted in the earlier parts of the articles together with the literature review. In my opinion lack conclusions or part of discussion is rather conclusions and summary. Paper needs some sentences on how your data will impact your field, or may be what you will do next to continouation of this.
References - literature cited is relevant, consistent with current knowledge.
The content of a paper is clear and easy to follow and presented an advance in current knowledge.
I think, the paper will be attract a wide readership not only for lichenologist.
Round 2
Reviewer 1 Report
Dear authors, I really appreciated the proposed integrations. I think that now the article is more complete, getting closer and closer to publication.
I found just a few points to bring to your attention.
Line 58: aliments
Line 79-97: wrong writing style
Line 110: check this period "Several species of Lobaria (Schreb.) Hoffm. is a frequently"
Line 127: put Escherichia coli in italic
Line 169-172: period to rewrite.
Line 216: "However,"
Line 221-239: It this part written using the correct font?
Line 229-230: check lichen/lichens
Line 231: I think this part "when humans use lichens" should be removed.
Reviewer 2 Report
In my opinion the manuscript has been sufficiently improved to warrant publication in Diversity. I am glad that my review and comments contributed to the improvement of this very interesting and valuable article. I fully accept all additions and improvements.
